# Interactions between Sleep and Emotions in Humans and Animal Models

**DOI:** 10.3390/medicina58020274

**Published:** 2022-02-11

**Authors:** Radu Lefter, Roxana Oana Cojocariu, Alin Ciobica, Ioana-Miruna Balmus, Ioannis Mavroudis, Anna Kis

**Affiliations:** 1Center of Biomedical Research, Romanian Academy, Iasi Branch, B dul Carol I, no. 8, 700506 Iasi, Romania; radu_lefter@yahoo.com; 2Department of Biology, Faculty of Biology, Alexandru Ioan Cuza University, B dul Carol I, no 11, 700506 Iasi, Romania; roxana_20_2006@yahoo.com; 3Center of Biomedical Research, Romanian Academy, B dul Carol I, no 8, 700505 Iasi, Romania; 4Academy of Romanian Scientists, Splaiul Independentei nr. 54, Sector 5, 050094 Bucuresti, Romania; 5Department of Exact Sciences and Natural Sciences, Institute of Interdisciplinary Research, Alexandru Ioan Cuza University of Iasi, Alexandru Lapusneanu Street, no. 26, 700057 Iasi, Romania; 6Department of Neurology, Leeds Teaching Hospitals NHS Trust, Leeds LS2 9JT, UK; i.mavroudis@nhs.net; 7Institute of Cognitive Neuroscience and Psychology, Hungarian Academy of Sciences, 1117 Budapest, Hungary; kisanna12@gmail.com

**Keywords:** sleep, emotions, human, animal models

## Abstract

Recently, increased interest and efforts were observed in describing the possible interaction between sleep and emotions. Human and animal model studies addressed the implication of both sleep patterns and emotional processing in neurophysiology and neuropathology in suggesting a bidirectional interaction intimately modulated by complex mechanisms and factors. In this context, we aimed to discuss recent evidence and possible mechanisms implicated in this interaction, as provided by both human and animal models in studies. In addition, considering the affective component of brain physiological patterns, we aimed to find reasonable evidence in describing the two-way association between comorbid sleep impairments and psychiatric disorders. The main scientific literature databases (PubMed/Medline, Web of Science) were screened with keyword combinations for relevant content taking into consideration only English written papers and the inclusion and exclusion criteria, according to PRISMA guidelines. We found that a strong modulatory interaction between sleep processes and emotional states resides on the activity of several key brain structures, such as the amygdala, prefrontal cortex, hippocampus, and brainstem nuclei. In addition, evidence suggested that physiologically and behaviorally related mechanisms of sleep are intimately interacting with emotional perception and processing which could advise the key role of sleep in the unconscious character of emotional processes. However, further studies are needed to explain and correlate the functional analysis with causative and protective factors of sleep impairments and negative emotional modulation on neurophysiologic processing, mental health, and clinical contexts.

## 1. Introduction

Being the most important functional restore and recovery process, sleep is described as a vital and cyclic mechanism occurring in most of the animal species. Despite that in the less complex organized species sleep features and chronology could be elementary, in mammals sleep follows a daily circadian pattern [1,2]. In general lines, sleep chronology includes five stages of which some are recurrent during the full sleeping time [3,4]. In humans, non-REM and REM sleep provide the physiological and behavioral context for activation of the mechanisms consequent to repair and restoration of optimal body functioning [5]. Learning and memory consolidation, cortical plasticity modulation, and cortical response potentiation are a few of the main neurophysiological effects of optimal sleep [6]. Furthermore, from a psychological point of view, during the sleep stages the emotional processing of emotionally perceived experiences’ intensity and emotional impact are provided by the neuromodulation of the emotionally significant salient memories encoded throughout the day [7]. Consistent with that, the disturbing emotional events could also modulate sleep duration and quality leading to mood and emotional state regulation [8]. Thus, an immediate detrimental effect of poor or negative sleep modulation could reverberate in altered physiological, psychological, and even pathological states. In this context, multiple correlations and a possible bidirectional cause–effect relationship between sleep regulation and psychiatric disorders were previously suggested [9,10]. Moreover, while the altered emotional processing was previously described in sleep deprivation [11], the emotional effects’ extent and the possible emotion-modulating sleep mechanisms have not been fully understood.

Thus, in this complex and bidirectional context of sleep–emotions modulation, it was our goal to find evidence and further describe the possible mechanisms behind the interaction between sleep and emotions’ perception and consolidation in both human and animal models, considering across species similarities and differences. In addition, considering the affective component of brain physiological patterns and the important impact of stress exposure on mood and sleep modulation, we aimed to find reasonable evidence in describing the two-way association between comorbid sleep impairments and psychiatric disorders.

## 2. Methodology

A thorough literature search was conducted mainly using PubMed and Web of Knowledge databases for articles concerning sleep and emotions physiology, mechanisms, regulation, and impairments, published from 2000 to present. Keywords (according to NLM controlled vocabulary thesaurus, MeSH) such as “sleep”, “emotion”, “sleep stages”, “sleep modulation”, “sleep mechanisms”, “REM sleep”, “non-REM sleep”, “slow wave sleep”, “emotions consolidation”, “sleep impairments”, “positive emotional response”, “negative emotional response”, “emotional memory”, “affective disorders”, “anxiety”, “depression”, “animal models”, “insomnia”, “amygdala”, and combinations were used to screen the mentioned databases for relevant content: title, abstract, and full content, respectively, according to PRISMA guidelines. Only English language written studies were considered for evaluation. Pre-screening and screening selections removed duplicate studies, foreign language studies, irrelevant studies, and studies for which updated research was available, according to PRISMA flowchart guidelines. Database screening was individually performed by three researchers, then the other three cross-checked the inquiry against inclusion and exclusion criteria. Any differences in opinion were solved by common consent.

## 3. Impact of Sleep on Emotional Dynamics

Regarding the physiology of sleep, many researchers agreed that sleeping is a complex process involving two main stages: the non-rapid eye movement (non-REM) classically divided into four stages and the rapid eye movement (REM) [12,13]. Several studies showed that due to the fact that the REM stage is characterized by increased cholinergic activity and also lacks aminergic activity, in this stage the processing of emotional memories occurs [12,14,15]. In this way, during REM sleep the anxiety-promoting neurotransmitters activity, such as serotonin and noradrenalin, is inhibited and the anxiety and fear experienced during the day could be reduced.

On the other hand, Tempesta et al. 2015 showed that sleep-deprived individuals could display negative evaluation of neutral stimuli rather than neutral or positive, in the recall phase of emotional memory tasks while assessing recognition accuracy for emotional pictures [16]. However, REM sleep deprivation was described to increase brain reactivity to negative and threatening visual stimulation [17]. Thus, the emotional processing areas (such as ventrolateral prefrontal, occipital, and temporal cortical areas) reactivity was significantly increased following one night of REM-deprived sleep, but not after non-REM deprivation [17]. In addition, considering that differences between early and late-night REM sleep were previously described [18], Khan et al. 2013 [11] suggested that the negative valence ratings of visual stimuli could be observed in late-night REM sleep, but not in early REM sleep or wakefulness. However, this correlation could be just the tip of the iceberg since the observed were more complex associations than previously predicted. Partial REM sleep-deprived individuals showed better capacity to emotionally adapt to negative stimuli in direct correlation to the time spent in REM sleep, as compared to healthy individuals [19]. These findings suggested that REM sleep deprivation could be directly correlated to negative emotional visual stimuli on arousal adaptability [19], thus it seems that the coping mechanisms underlying emotional balance gain more relevance directly dependent on the amount of stress to which the individual is exposed.

Another important mechanism which is thought to be characteristic of REM sleep could be the protection of the emotional response to the sight of negative images, otherwise reduced in the wakefulness state [20]. Moreover, it was suggested that the consolidation of negative versus neutral stimuli could be better observed when sleep-deprived subjects are selectively deprived of late-night REM sleep rather than of early-night slow-wave sleep (SWS) in the non-REM stage [12,13,21]. In this context, the fact that the changes occurring in the delta frequencies of REM and SWS following one-night partial sleep deprivation could be correlated to self-reported positive effects’ inhibition rather than negative affective processes’ facilitation [22] could bring additional evidence of the mechanisms that correlate the sleep patterns and the emotional perception in a complex interaction. However, an important consequence of selectively depriving REM and non-REM SWS, according to Weissner et al. [21], was that none of the two stages-deprived individuals reported self-perceived changes in mood, while the authors pointed out that their endeavor brings further knowledge on the “sleep to remember-sleep to forget” hypothesis of memory consolidation.

Yet, Menz et al. 2016 [23] showed that fear conditioning correlated stimuli extinction in the context of threatening and non-threatening stimuli discrimination could be a late-night REM-rich sleep feature rather than a SWS-rich one. Additionally, they showed that the proportional correlations between REM sleep amount and autonomic responses [24] suggest that REM sleep is involved in both fear conditioning learning and fear extinction/negative emotional release during the night, these mechanisms being essential in preventing anxiety disorders, such as phobias [25].

In the context of REM sleep impairment effects, REM sleep has also been correlated to depressive behaviors, with altered REM latency (shortened) and density (increased) being currently used as sleep physiology-related markers in depression diagnosis [25]. However, a bidirectional cause–effect relationship between sleep and depressive behavior was previously documented by Franzen and Buysse 2008 [26], Pasquale et al. 2013 [27], and Muphy and Peterson, 2015 [28]. Considering this bidirectional modulation between sleep deprivation (insomnia) and depressive behaviors, it was suggested that differentiating the cause-and-effect relationships is rather difficult and remains inconclusive.

In this way, despite the fact that the sleep patterns in animal models could greatly vary, as compared to humans, animal studies replicating the effects of sleep deprivation in humans are based on neuroimaging and multiple physiological parameters (body temperature, respiration rate, brain, and blood chemistry) [29]. As a result, some animal model studies managed to describe a correlation between REM sleep deprivation and decreased emotional learning. Thus, it was proven that in a paradigm using electrical foot shock fear conditioning learning in rats, REM sleep deprivation or sleep fragmentation could lead to impaired extinction of conditioned fear as well as long-term fear memory deficits [30,31]. Consequent to the cause–effect bidirectional relationship previously discussed in humans, Wellman et al. 2017 [32] showed that fear conditioning and fearful context re-exposure altered subsequent REM sleep in Wistar rats [32].

Thus, the shortened REM sleep effects on emotional reactivity could be related to the oscillation between cortical and subcortical structures for a specific threat-relevant stimulus appraisal. During this sleep phase, the amygdala appears to exert a preferential neuromodulatory influence on emotional over neutral information in the hippocampus, mediated by the rhythmic release of the stress hormone cortisol, which correlates with the REM sleep episodes [33,34]. In a study aiming at the pre-surgical evaluation of epileptic patients, Corsi-Cabrera et al. 2016 [35] showed that amygdala activity was associated with the rapid eye movements during the REM sleep, using stereotaxically implanted electrodes to record brain activity. These results suggested that amygdala could be an essential component of REM sleep physiology exhibiting a direct excitatory role associated with the rapid eye movements [35] and further confirming prior results observed only in animal studies.

Similarly, the fact that anxiety-like behavior observed after acute 24 h of REM sleep deprivation in rats was reduced after the central administration of neuropeptide S (commonly used to normalize the sleep-wake pattern and subsequent sleep rebound) [36] could further confirm the simultaneous and interactive activation of REM sleep and emotional processing brain structures. Additional research on this observation revealed that neuropeptide S determined neuronal depolarization in multiple regions of the amygdala, but mainly in the excitatory basolateral axonal projections on the central nucleus of amygdala which is implicated in pain processing and, in this case, in mediating the anxiolytic behavioral effects related to sleep disturbances [36].

Further evidence regarding the joint activity of amygdala in both emotions processing and sleep was brought by several recent studies showing that many brain regions which are involved in REM sleep regulation could also be upregulated by amygdala hyperactivity (occurring in anxiety, fear, and potentially threatening ambiguous stimuli processing) [37,38,39,40,41,42,43]. In this way, the superior cortical regions, the medial prefrontal cortex including Brodmann areas 24 and 32 in the anterior cingulate, the caudal orbitofrontal regions, the insula, the entorhinal cortex, and para hippocampal cortex [37,38,39,40,41] are all involved in emotional memory processing, receive input from the amygdala, and are important modulators of the REM sleep phase [42]. Repeated functional magnetic resonance imaging scans and electroencephalographs that recorded brain activity over a sleepless night showed increased bilateral reactivity of the amygdala, associated with changes in the ventromedial prefrontal cortex functional connectivity during the emotional stimuli rating task [43]. Furthermore, it was suggested that overnight wake decreased behavioral reactivity in correlation with the reduced REM sleep gamma waves and decreased central adrenergic activity [43].

On the other hand, in animal models, it was demonstrated that the consolidation of fear memory following stressful fear conditioning is induced by the increased theta waves frequency synchronization between the hippocampal (CA1) region, the amygdala (lateral nucleus), and the medial prefrontal cortex, which generally occurs during the late-REM phase of sleep while increased levels of corticosterone are released [44]. Moreover, mice emotional memory consolidation impairment could occur following the inhibition of the medial septum GABA neurons, which project to the hippocampus and normally pace the hippocampal theta rhythm during REM sleep [45]. In addition, the inactivation of the basolateral amygdala activity in Wistar rats with the GABA agonist muscimol, prior to fear conditioning and polysomnography recording, did not alter REM sleep duration, as compared to untreated rats [32]. However, in a place-threat association task performed on a rat model, the coordinated reactivation of dorsal hippocampal neurons and a preferentially upmodulated subgroup of basolateral amygdala cells were signaled during non-REM sleep, suggesting that this could also be a contextual emotional memory consolidation mechanism [46]. In this way, it could be suggested that both REM and non-REM sleep could be actively implicated in the emotional memory consolidation and emotional relief concurrent with the activity of amygdala structures, as described by the recent animal model studies. This aspect could be important evidence of the fact that sleep does not bring only physical benefits to the individual, but also daily accumulated neurophysiological stress relief, as previously hypothesized.

Moreover, while the REM phase is characterized by increased cholinergic status and two-fold acetylcholine levels during quiet waking [47], the stress modulating central adrenergic neurotransmission remains almost completely inhibited [48]. Similarly, brainstem acetylcholine activity could stimulate some REM sleep active brain structures of the forebrain and cerebral cortex, which are also encouraging alertness and wakefulness [49]. Taken together, these findings could suggest that the REM sleep phase promoted the reprocessing of previous emotional experiences by reducing their intensity and the release of affective brain circuits of the neurohumoral load which is thought to prevent anxiety disorders-derived manifestations [25]. In this context, large cross-sectional studies and also in-depth laboratory investigations described the significant correlations between sleep parameters and long-term emotional state or emotion-related personality traits. Thereby, a Korean population-based prospective cross-sectional investigation [50] of 2404 men and 2291 women and a questionnaire survey of 5433 hospital employees [51] found that the presence of increased anger was significantly correlated with sleep impairments and its decreased quality (including non-restorative feelings). Furthermore, a cross-sectional web survey of 5129 adult subjects [52] also showed that positive emotional traits, such as high control and drive, correlated with better sleep profiles, while increased anger was significantly associated with poor sleep.

Thus, it seems that the affective influence on sleep is a two-way road, since numerous studies manage to describe the bidirectional relationship between sleep and emotional response. Better sleep quality consistently predicted a next-day higher positive affect and a lower negative affect was described in the multilevel model within-person studies [53,54] on daily experiences’ association with sleep quality and duration in adult employees. These associations were also reported in adolescents: negative valence effects, such as family-related stress and negative social emotions (e.g., rejection, anxiety) were strongly related to lower self-reported sleep efficiency, but not sleep duration [55], or to both [56]. Interestingly, low arousal affective experiences predicted more favorable sleep outcomes, while high arousal emotions led to longer sleep latency, regardless of their valence. Thus, dysphoria and calmness can lead to higher sleep efficiency; however, this could be caused by different mechanisms, a coping strategy being longer sleep in the case of dysphoria, in contrast to a sleep-inducing protective environment being ensured by calmness [56].

## 4. Mechanisms Involved in the Relationship between Sleep and Emotions

The amygdala is directly responsible for associating emotional significance for the received information, as well as storing, coding, and recalling emotional memories. In addition, it is related to the prefrontal cortex, which is involved in working memory, motivation, planning, and diminishing fear reactions. However, it also responds to the release of neurotransmitters which play an important role in neuronal activation: dopamine, noradrenaline, serotonin, histamine, and hypocretin/orexin [57].

Furthermore, the amygdala is also involved in sleep regulation mechanisms. Narcolepsy with cataplexy is a particularly salient example. The loss of orexin cells results in narcolepsy, while cataplexy is the amalgam of wakefulness and REM-like atonia [58]. The neural cause of cataplexy is the loss of hypocretin connections to the locus coeruleus, and the second largest connections are to the amygdala. The most salient triggers for human cataplexy are laughter and sudden pleasant emotions, that is, following this positive emotion, REM atonia occurs [59], revealing a direct link between emotions and the neural structures of sleep. The orexin neuropeptides (orexin A and B) produced in the hypothalamus are also known to promote wakefulness and arousal through their excitatory projections to all monoaminergic cell groups [60]; they have also been found to promote facilitating antidepressant effects and positive drive [61]. Interestingly, a blockade of both orexin (OX1 and OX2) receptors by selective antagonists that was identified as a common molecular mechanism across mammalian species, identified in mice, rats, dogs, and humans, was found to promote sleep and increase both REM and non-REM sleep [62], even without inducing the lack of orexin signaling associated with cataplexy [63]. Furthermore, bilateral microinjection of orexin A into the central amygdala, a densely orexinergic innervated region, modulated the spontaneous firing activity of central amygdala neurons and induced significantly anxiolytic-like behaviors in rats [64]. Moreover, in recent studies, orexin receptors were found to play a role in the acquisition and extinction of conditioned fear, as microinjections in the amygdala of OX1 receptor antagonists or genetic knockout of OX1 receptors significantly decreased fear memory acquisition and decreased sensitized fear [65,66].

Serotonin, which exerts inhibitory control over REM sleep, mainly via activation of serotonin1A receptors [47], may also mediate mood changes. Selective serotonin reuptake inhibitors’ administration was associated with reduced REM sleep [67]. Moreover, depressive-like symptoms observed in rats after chronic constant light exposure for 6 weeks were reversed by agomelatine, a melatonin receptor agonist and serotonin 2C receptor antagonist [68]. Excessive extracellular serotonin load at an early age in the brain of knockout mice lacking the serotonin transporter generated age depressive-like behaviors and altered REM sleep duration in adults [69]. An interesting interplay between orexin, serotonin, and prolactin was suggested to mediate sleep and narcolepsy [70], and a recent study identified the molecular mechanism of direct serotonergic regulation of orexin neurons in mice [71].

During sleep deprivation, a diminished functional connectivity between the amygdala and the anterior ventral cingulate cortex may alter the perception of external stimuli or even the tendency of responses to negative stimuli [72]. In individuals with a poorer sleep quality, amygdala reactivity was direct proportional to the extent of depressive symptoms and perceived psychological stress [73]. REM sleep deprivation reduces inhibitory control exercised by the medial prefrontal cortex to suppress amygdala activity, whereas other circuits, such as those between the amygdala and brainstem autonomic regions (from the pre-limbic prefrontal cortex to the paraventricular nucleus of the thalamus), become activated contributing to emotional instability and exacerbating fear memory [30,74]. Increased anxiety-like phenotypes appearing in juvenile male rats after prolonged REM sleep restriction are accompanied by higher noradrenaline levels in the amygdala and ventral hippocampus [75]. Similarly, elevated noradrenaline levels and anxiety-like behavior observed in sleep-deprived zebrafish were reversed by antagonist action on a1-adrenoceptors, which also increased the sleep period [76]. It was suggested that under REM sleep deprivation conditions, the elevated noradrenaline level in the brain acting on a1-adrenoceptors would reduce membrane lipid peroxidation, which would further modulate intracellular ionic concentrations and thereby increase Na-K-ATPase activity and neuronal excitability [77].

Furthermore, evidence showed that sleep impairment could also alter fear/emotional memory consolidation through the cyclic adenosine monophosphate (cAMP) and its related transcriptional pathway (CREB) which has been suggested to intensify during REM sleep [78]. In this way, in a study regarding the alleviation of sleep deprivation-induced deficits in long-term fear memory it was demonstrated that under REM sleep deprivation conditions, fear memory deficits observed in rats were accompanied by decreases in CREB proteins and increased catalysis of cAMP in the hippocampus and striatum [31], suggesting that REM sleep plays an important role in the consolidation of long-term fear memory through mechanisms localized in the hippocampus, striatum, and amygdala.

The habenula, an epithalamic bilateral structure located above the thalamus, has been described as a component of the neural system regulating circadian rhythmicity [79]. Moreover, the lateral habenula has been suggested to regulate REM sleep by maintaining the specific hippocampal theta rhythmicity via serotonergic projections to the raphe nuclei and thereon to the hippocampal interneurons [80]. The habenula is also likely to be implicated in sleep–mood regulation, as it becomes hyperactive during REM sleep fragmentation and mediates depression-like symptoms in rats [81]. An increased firing rate of cholinergic neurons in the habenula of selectively REM sleep-deprived mice was correlated with cell-autonomous mechanisms that were independent of synaptic transmissions, such as the reduced activity of TASK-3 K+ channels, which were sensitive to changes in intracellular K+ following sleep fragmentation [82].

Selective REM sleep deprivation in rats caused a progressively increased homeostatic drive for REM sleep that correlated with increased brain-derived neurotrophic factor (BDNF) protein expression in the pedunculopontine tegmentum and nucleus subcoeruleus [83], important areas that regulate REM sleep [84]. Guo et al. 2016 found that 6 h of REM sleep deprivation in rats did not affect BDNF in the prefrontal cortex, but induced a decrease in BDNF in the hippocampus, striatum, and amygdala, disrupting long-term fear memory [31]. Moreover, early age stress caused by cross-fostering in Wistar rat pups, which altered the REM sleep onset at adulthood and increased the duration during the light period, also lowered the BDNF expression in the basal forebrain, which may reflect cholinergic neurotransmission defects and suggest that this is a risk factor for developing later cognitive impairment [85].

Slow-wave sleep inhibits the hypothalamic–pituitary–adrenal axis (HPA) axis and cortisol secretion [86]. The intracerebroventricular administration of the corticotropin-releasing hormone [87,88] or systemic administration of glucocorticoids [89] could lead to arousal and sleeplessness. Plasma cortisol elevated levels, characteristic for wakefulness and stage 1 sleep, but not for late-sleep stages, accompany low quality sleep and insomnia and the subsequent physical and mental tiredness [90,91] which are correlated to altered emotional processing [92,93]. Deregulation of HPA axis associated with chronic insomnia might be an important component linking inadequate sleep to stress-related pathology [94]. It should be noted that with chronic lack of sleep, progressive increases in glucocorticoid levels may occur, which may shift the brain from initial adaptation to disease, which could impair neuronal neurogenesis and plasticity. This could unequivocally lead to behavioral disorders and pathophysiological mechanisms of mood disorders [94].

## 5. The Social Context of Sleep and Its Relationship with Emotions

Emotions possess an essential role in regulating social interaction, motivating and preparing physiological reactions, motor behaviors, and behavioral decisions [95]. Additionally, it was suggested that emotions could interfere in internal processes such as perception, attention, learning, memory, and motivational priorities [95]. In this context, sleep seems an important determinant of social dynamics, as emotional processing appears to be highly sensitive to even temporary sleep disruptions that impair the affective neural systems [96]. One night of acute sleep deprivation in healthy adults significantly increased anxiety state, anhedonic depression, and general distress (using common metric self-reported anxiety scales) [97]. Daut et al. 2019, using a complex paradigm of chronic earlier light onsets to replicate the modern society circadian disruption in rats, reported an increased susceptibility to depression-related manifestations during stress exposure, reminiscent of the greater prevalence of mood alterations in night shift workers [98]. Recently, a consistent number of studies investigating the effects of sleep and specifically sleep loss acknowledged that the quality of social relationships is related to sleep patterns. Increased impulsivity to negative emotions or greater negative affective behavior occurring in peer social contexts were reported following sleep deprivation under laboratory conditions [99,100].

Similarly, increased social risk behavior (e.g., rebelliousness, drug use) was reported to be significantly associated with a reduced amount of sleep, under the optimal 8–9 h (specifically under 7 h) in a large survey data analysis on the daily impact of sleep conducted for two separate years on over 10,000 eighth-grade students [101]. Surprisingly, a longer than optimal sleep duration, over 10 h, was also associated with increased behavioral risk factors [102]. Lower subjective sleep quality experienced in daily life, measured by actigraphy and questionnaires, was significantly associated with lower empathic ability in healthy young adults in computerized emotional empathy tasks [102].

An assessment of a partial or total night of sleep deprivation under laboratory conditions indicates that sleep quantity and sleep quality are an important determinant for social interaction, influencing emotional decoding, sharing, and communication. Thus, besides the vastly cited impairing of human facial emotions’ assessment, attenuated emotional expressiveness was also reported, with significantly slower facial reactions in response to emotional stimuli recorded by facial electromyograms [16] or by human observers [103]. Pitch decreased and less intense vocal emotional expressiveness was detected after one night of sleep deprivation by special computerized software and was particularly pronounced in adolescents relative to adults [104]. This reduced emotional involvement was noticeable in relation to positive stimuli, such as amusing film clips versus sad film clips or a decreased number of positive emotional words versus an unmodified number of negative emotion words when conveying a verbal message [103,104].

Furthermore, total sleep deprivation for one night was reported to cause an overall emotional blunting by significantly reducing empathic reactions to the emotional content of standardized affective pictures [105]. A highly evident yet less regarded aspect concerns the change in looks caused by lack of sleep, even for only 4 h. The inherent reduced blood flow to the skin and increased paleness of the face could lead to decreased attractiveness and suggests health risks in the surrounding social environment. Observers of photos of healthy adults subjected to two days of acute sleep deprivation “were less inclined to socialize with”, butnot saw them as less trustworthy [106].

However, it could be relevant to note the predominance of negative emotions in the current literature, while the positive emotions’ effects are merely discussed. In this way, a recent systematic review [107] having a similar hypothesis concluded that the need for further analysis is critical, since a few unbiased studies could support the positive emotions’ effect on sleep in healthy individuals while limited empirical data were not sufficient to provide reasonable evidence for positive emotional outcomes in clinical population. Later studies managed to show that positive affect increased sleep quality [108,109,110], improved self-control, and prevented mood changes [111], in healthy individuals and chronic non-psychiatric (e.g., thalassemia [112], psoriasis [113]) and psychiatric pathologies [114,115]. By exception, Bouwmans et al. [116] reported no correlation between positive affect and sleep quality in depressed patients. Currently, it is increasingly accepted that a good welfare is not merely the lack of negative emotions, but the presence of positive emotions. In this context, from the above discussed studies only Marakkemia et al. [113] evaluated and highlighted the possible beneficial effects of encouraging positive emotions.

## 6. Interaction between Sleep Quality, Mental Health, and Psychiatric Disorders

Generally, continuous wakefulness for 1–2 nights is shown to decrease self-reported positive mood [117,118,119] and worsen self-reported negative mood states (i.e., anger, depression, anxiety, confusion, and paranoia) coupled with somatic complaints and fatigue [120,121]. One night of sleep deprivation assessed in a cohort of university students led to more negative perceptions of neutral stimuli, increased negative mood, and decreased subjective alertness [122]. These data are suggestive in the context of the effects of exertion placed on the nervous system by even shorter periods of lack of sleep. Similarly, animal vulnerability to sleep impairments and altered behavioral traits were previously demonstrated. For example, in dogs, emotionally positively or negatively loaded interactions markedly and differently changed sleep physiology, including REM sleep duration and sleep latency [123]. In addition, chronic constant light exposure for 6 weeks in rats or only 6 h of total sleep deprivation in mice caused significantly increased depressive-like symptoms [68,124]. Furthermore, sleep disturbances were also accompanied at the level of the prefrontal cortex by increased oxidative stress and GABAergic inhibition in the supra-solicited parvalbumin neuronal cells in rats, features suggested to underlie some of the cognitive deficits found in schizophrenia sleep deprivation phases [125]. Moreover, schizophrenia-like behavioral phenotypes, including fear memory impairments, were observed in mice exhibiting increased wakefulness at the expense of non-REM sleep after genetic ablation of serotonin 2B receptors [126]. Thus, consequent to these animal models studies, it seems that sleep quality and duration is correlated not only to mood states, but also to some affective or non-affective neuropsychiatric disorders. In this way, one could suggest that despite the fact that the cause–effect relationship between sleep and psychiatric impairment remains unclear, the mechanisms underlying both mood and emotional states and sleep regulation comprehensively interact.

In order to bring further evidence on these aspects, recent studies showed that sleep and dreaming are vital functions for the maintenance of “psychic homeostasis” [127], while the impact of sleep deprivation leads to aggravating outcomes during chronicization. Thus, chronic insomnia (characterized by psychologically incapacitating sleep deprivation) is the most common comorbidity associated with psychiatric disorders [128]. Furthermore, insomnia’s high prevalence was reported in older adults due to the partial decline in the functionality of sleep control systems with age, while in women this is due to the onset of menses and menopause [129]. Insomnia also affected adolescents and younger age groups and was strongly associated which everyday use of different electronic media of over 3 h/day [130]. The outcomes of insomnia reverberate across the emotional and social dimensions: a study on 366 adolescents based on self-reported questionnaires related to emotions, empathy, and sleep found a significant correlation between subjective insomnia and specific impairments in emotional competence and empathy leading to decreased quality of life [131].

In addition, patients with chronic psychophysiological insomnia or sleep apnea syndrome showed a significantly lower performance in identifying basic facial emotion expressions [132], tended to rate facial expressions as less emotionally intense than healthy individuals, and were prone to self-reported anxiety and depression [133]. Thus, chronic insomnia was associated with up to 40% of cases of severe comorbid medical and psychiatric conditions, such as depression or anxiety [91]. While being directly correlated to altered mood, affective disorders were also reported to be associated with emotional distress and were markedly more present in individuals exhibiting anxiety-like emotion-related personality traits [134].

In unipolar disorder, chronic sleep impairments increased the negative affect towards unpleasant events, as compared to healthy individuals in which these effects had a lesser extent [135]. On the other hand, in bipolar disorder sleep impairments are prone to occur particularly during acute mood episodes [136,137]. In this way, the importance of sleep quality and the direct correlation between better sleep and better recovery were demonstrated in a study examining the sleep–wake cycle by wrist actigraphy in adolescents diagnosed with a borderline personality disorder or bipolar disorder during the euthymic phase [138]. The fact that patients reported 1 hour-longer sleep time, as compared to healthy individuals, could suggest this positive outcome and also the fact that affective impaired individuals need a longer period of time to recover in relation to emotional processing and negative emotional release during sleep, as previously described.

Regarding the negative emotional release during sleep, the low distress tolerance of negative emotions being inappropriately perceived as uncontrollable, threatening, and intolerable was observed [139,140]. While this impairment was often reported in many forms of psychopathology, including depression, anxiety, post-traumatic stress disorder, and substance abuse, some studies suggested a strong correlation between poor sleep quality and low distress tolerance in the mentioned pathological conditions [139,140]. Furthermore, concurring with the bidirectional cause–effect relationship between sleep and mood hypothesis, several recent studies showed that co-occurring sleep impairments in mood disorders could also lead to emotion regulation difficulties and exacerbated negative consequences in cognitive reappraisal [141,142,143,144,145,146].

While chronic sleep impairments were shown to increase negative emotional reactivity and to play a major role in psychiatric conditions, the sleep disorder’s management could lead to negative emotional and altered affective symptoms relief. In this context, Chu et al. 2015 [147] showed that sleep improvement and emotional processing (evaluated by facial emotion recognition tasks) were concomitantly occurring in psychiatric disorders patients, while the lack of sleep was significantly associated with the incapacity to recognize angry and fearful expressions, specifically. This could be the reason why sleep quality and duration were reported the best predictors of better outcomes in treating anxiety disorder, as a study on 11-year-old children recently showed [148].

Considering these aspects, it could be suggested that not only a strong correlation between sleep, mental health, and emotions could be observed in both patients and animal studies, but also that the pathological mechanisms leading to impaired sleep are significantly interacting with the pathological mechanisms which determine many of the psychiatric disorders (mainly the affective spectrum disorders, but not restricted to them). However, the direction in which the interactions occur is not yet fully understood and most of the studies rely on bidirectional modulation.

## 7. Interaction of Sleep, Emotion, Age, and Gender

In the context of physiological changes that could lead to sleep and emotional processing differences, evidence showed that sleep deprivation effects are more aggressive in adolescents, as compared to adults and elders. This observation could be argued by the fact that during this age the individual crosses paths with an intense brain remodeling process which could predispose them to increased vulnerability to social-affective stress stimuli processing impairments [149]. Thus, the lack of sleep at this age could lead to an altered prefrontal cortex inhibitory effect on the amygdala. Other sleep features, such as increased sleeping time (10 h per night) or an up to 40% decrease in SWS phase-time [101], could also lead to impaired emotional states in adolescents and youngsters.

The studies on animal models regarding sleep patterns and effects on brain functioning showed that sleep deprivation at an early age is detrimental to emotional development [150]. In this way, third-trimester pregnant dams had REM sleep deprivation for 5 h, which caused increased distress vocalization after birth and juvenile anxiety-related behaviors, suggestive of emotional instability in newborns [150]. In juvenile male rats, REM deprivation caused higher basal corticosterone levels, impaired physical development, and anxious phenotypes [75]. Conversely, emotional stress in rats consisting of postnatal maternal separation and isolation stress lead to increased REM sleep and total time, while increased fear memory retention and fear generalization following differential fear conditioning were notable [44].

Considering the demonstrated physiological, psychological, and emotional contrasts between genders, several differences in sleep and sleep impairments’ interactions with the mentioned traits could be observed as function to gender. In this way, in women, sleep impairments were more often associated with the risk of developing hypertension, cardiovascular, and metabolic disorders, as compared to men [151]. A recent study on the possible 36-h sleep deprivation effects in healthy adolescents reported that affective states were significantly worsened, but only female participants reported anxiety and depression alongside anger, confusion, and fatigue [120]. In addition, the fact that salivary cortisol levels were reported as significantly increased in young women, but not in men, as a result of sleep deprivation [152] could suggest that HPA axis stress response modulation mechanisms which interact with sleep modulatory patterns are differentially activated in relation with the individual gender. Furthermore, the presence of sleep and mental disorders as comorbidity to chronic pathological conditions was significantly prevalent in women, as compared to men of a large demographic study on 850 hospitalized chronic obstructive pulmonary disease patients [153]. Facial emotions perception impairments were also reported in one-night sleep deprivation performed on women, mainly regarding the threat-relevant and reward-relevant facial mid-intensity range emotions (anger and happiness, respectively) [96]. By contrast, some studies failed to find gender differences in basic emotions’ recognition accuracy [154,155].

However, some of the mentioned gender differences could be explained by the different susceptibility to circadian rhythms and hormonal changes during menstrual cycles. Women were found to have a significantly shorter circadian period, with higher phase-advanced endogenous temperature and melatonin rhythms, which contributed to a higher prevalence of insomnia and sleep quality problems [156]. In this context, it was also suggested that sleep impairments in both women and men could also lead to fertility disturbances and vice versa, as showed by the authors in [157,158,159,160]. Furthermore, clinical studies on humans and rodents show that sleep phases are strongly modulated during the menstrual cycle by several mechanisms involving high progesterone and estradiol concentrations in the direction of decreased REM sleep and increased non-REM sleep times [151]. From an evolutionary point of view, the gender-related sleep differences could be linked to adaptative neurological function-based processes necessary to facilitate species preservation. According to Ferrara et al. 2015, one night of sleep deprivation could alter decision-making behavior and social preference in a gender-sensitive way, causing females to exhibit a heightened aversion to risky choices and become less altruistic, which would be beneficial for offspring survival [161].

## 8. Conclusions

A strong modulatory interaction between sleep processes and emotional states and processing was observed in both physiological and pathological conditions, as demonstrated by human and animal model studies. The fact that sleep quality and duration could significantly modulate emotional states and mental health suggested that physiologically and behaviorally related mechanisms of sleep are intimately interacting with emotional perception and processing, with most of the conclusive findings being provided by human studies and also animal model studies (in a scarce species variety). The hypothesis of bidirectional modulation between sleep mechanisms and emotional processing has recently been more often addressed and described. However, further studies are needed to explain and correlate the functional analysis with causative and protective factors of sleep impairments and negative emotional modulation on neurophysiologic processing, mental health, and clinical contexts.

## Data Availability

Not applicable.

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
