# Peer review of "Interactions between Sleep and Emotions in Humans and Animal Models"

_medicina, 2022, doi:10.3390/medicina58020274_

Round 1

Reviewer 1 Report

Dear authors

Thanks for writing this article

Review studies are expected to find study gaps and provide practical suggestions in various fields for researchers and clinical professionals.
This review article lacks these features. In other words, it does not have points to previous review articles.

Author Response

Reviewer 1:

Dear authors,

Thanks for writing this article.

Review studies are expected to find study gaps and provide practical suggestions in various fields for researchers and clinical professionals.

This review article lacks these features. In other words, it does not have points to previous review articles.”

Response:

Dear Reviewer 1,

Thank you for your willingness to read our paper and for kind considerations which helped us to realize the limitations of our approach. Since our work was previously narrative rather than argumentative, we thoroughly analyzed now the presented ideas and further discussed them so that the previous studies’ findings to be weaved in our argumentation. Also, we managed to reformulate the way we presented our aim so that it could better express our research question. Please find attached the updated version of our manuscript in which all the changes were made using the “Track changes” option in MS Word.

Reviewer 2 Report

The Authors present an important issue of sleep affecting the mental condition and the other way round. The review is thorough and contains many, over one hundred, references.

My comments:

1/ The Authors did not follow PRISMA guidelines while preparing their review. I would recommend to add the methods section where they should describe how they performed their review, which databases and MeSH they used and if there were any excluding criteria.

2/ The cited references are quite old. Only 41 of them (out of 135) come from the last 5 years. I would suggest adding newer papers, similar to yours where this bidirectional relationship between sleep and mental health is presented, for instance:  DOI: 10.3389/fpsyt.2021.674460. You could add this to the paragraph where you mentioned this mutal relationship in different groups of subjects (lines from 151).

3/ Moreover, there are disturbances in the references - numbers are doubled - please correct, and references are not prepared in the style of MDPI requirements.

4/ In the paragraph about the reproduction and menstrual cycle (starting from 409) I would probably add one sentence about the negative influence of sleep disorders on the fertility.

Author Response

 Reviewer 2:

The Authors present an important issue of sleep affecting the mental condition and the other way round. The review is thorough and contains many, over one hundred, references.

My comments:

1/ The Authors did not follow PRISMA guidelines while preparing their review. I would recommend to add the methods section where they should describe how they performed their review, which databases and MeSH they used and if there were any excluding criteria.

2/ The cited references are quite old. Only 41 of them (out of 135) come from the last 5 years. I would suggest adding newer papers, similar to yours where this bidirectional relationship between sleep and mental health is presented, for instance: DOI: 10.3389/fpsyt.2021.674460. You could add this to the paragraph where you mentioned this mutal relationship in different groups of subjects (lines from 151).

3/ Moreover, there are disturbances in the references - numbers are doubled - please correct, and references are not prepared in the style of MDPI requirements.

4/ In the paragraph about the reproduction and menstrual cycle (starting from 409) I would probably add one sentence about the negative influence of sleep disorders on the fertility.”

Response:

Dear Reviewer 2,

Thank you for your patience to thoroughly read our work and for kind suggestions which helped us to improve the quality of our manuscript. Hereby, we detail the changes we made in our ms, addressing all your suggestions. Please find attached the updated version of our manuscript in which all the changes were made using the “Track changes” option in MS Word.

1/ The Authors did not follow PRISMA guidelines while preparing their review. I would recommend to add the methods section where they should describe how they performed their review, which databases and MeSH they used and if there were any excluding criteria.

We added a method section in which we described the way we performed our review, including the databases from where the studies were selected for screening and keywords (according to NLM controlled vocabulary thesaurus) we used to generate pre-screening lists of possible relevant studies. We followed the PRISMA guidelines, prepared a PRISMA workflow, and considered inclusion and exclusion criteria while constructing our argumentation, but presented it as a narrative review.

2/ The cited references are quite old. Only 41 of them (out of 135) come from the last 5 years. I would suggest adding newer papers, similar to yours where this bidirectional relationship between sleep and mental health is presented, for instance: DOI: 10.3389/fpsyt.2021.674460. You could add this to the paragraph where you mentioned this mutal relationship in different groups of subjects (lines from 151).

Considering that few referenced studies were newer than 2018, we conducted a new search following that year, and updated our argumentation according to the newer available data. Also, if newer study versions were available, they replaced the older references. We also added the mentioned work and several relevant others which addressed the bidirectional relationship between sleep, emotion processing, and mental health. We added some discussions on this aspect in some paragraphs throughout the manuscript.  

3/ Moreover, there are disturbances in the references - numbers are doubled - please correct, and references are not prepared in the style of MDPI requirements.

We reviewed, formatted, and updated the References section.

4/ In the paragraph about the reproduction and menstrual cycle (starting from 409) I would probably add one sentence about the negative influence of sleep disorders on the fertility.”

We added the suggested aspect in a short discussion at the mentioned lined (newly shifted to line 527).

Thank you !

Round 2

Reviewer 1 Report

This modified version is worth publishing.

This manuscript is a resubmission of an earlier submission. The following is a list of the peer review reports and author responses from that submission.